# Genetic dissection of clonal lineage relationships with hydroxytamoxifen liposomes

Ryan C. Ransom[1,2], Deshka S. Foster[1,2], Ankit Salhotra[1,2], Ruth Ellen Jones [1,2], Clement D. Marshall[1,2], Tripp Leavitt [1], Matthew P. Murphy[1,2], Alessandra L. Moore[1], Charles P. Blackshear[1], Elizabeth A. Brett[1], Derrick C. Wan[1] & Michael T. Longaker[1,2]

Targeted genetic dissection of tissues to identify precise cell populations has vast biological and therapeutic applications. Here we develop an approach, through the packaging and delivery of 4-hydroxytamoxifen liposomes (LiTMX), that enables localized induction of CreER$^{T2}$ recombinase in mice. Our method permits precise, in vivo, tissue-specific clonal analysis with both spatial and temporal control. This technology is effective using mice with both specific and ubiquitous Cre drivers in a variety of tissue types, under conditions of homeostasis and post-injury repair, and is highly efficient for lineage tracing and genetic analysis. This methodology is directly and immediately applicable to the developmental biology, stem cell biology and regenerative medicine, and cancer biology fields.

[1] Department of Surgery, Division of Plastic and Reconstructive Surgery, Stanford University School of Medicine, Stanford, CA 94305, USA. [2] Institute for Stem Cell Biology and Regenerative Medicine, Stanford University School of Medicine, Stanford, CA 94305, USA. Correspondence and requests for materials should be addressed to M.T.L. (email: longaker@stanford.edu)

The ability to genetically isolate and trace the clonal expansion and fate of individual stem and progenitor cells has wide-ranging applications in biology and medicine. The Cre/lox recombination system permits tissue-specific labeling[1] in conditional transgenic and gene knockout animals[2]. With fusion of the ligand-binding domain of the estrogen receptor (ER) to the Cre recombinase, temporal control over induction can be achieved as recombination only occurs after tamoxifen is administered and binds to the ER[3]. However, spatial control over induction remains limited to the cell and tissue-specific expression of Cre in available driver strains. With systemic tamoxifen administration, all tissues expressing the Cre driver are induced, which can be very broad depending on the driver selected. There is a potential lack of promoter specificity as well as a possible lack of validated promoters in certain cell types. A more refined system with temporal and spatial control over CreER[T2] activation is beneficial to analyze complex tissues.

Efforts have been made previously to use optical "uncaging" of CreER[T2] activity (using one and two-photon photo-activation) to specifically label cells of interest[4]. However, this technique requires delivery of caged compounds as well as light-mediated activation, which can be invasive and difficult to achieve in certain tissue types. Liposome-encapsulated tamoxifen has previously been explored for topical application in the setting of skin carcinogenesis in order to avoid the systemic anti-estrogen effects[5,6].

We devised an innovative strategy for the packaging and delivery of 4-hydroxytamoxifen to locally induce CreER[T2] recombinase in transgenic mice. This methodology is efficient for cell labeling using a variety of fluorescent labeling constructs in mice with specific or ubiquitous Cre drivers in multiple different tissue types and both at homeostasis and during repair post-injury.

## Results

**Generation of 4-hydroxytamoxifen liposomes**. We generated hydrophobic liposomal vesicles[7] and produced 4-hydroxytamoxifen liposomes (LiTMX) (Fig. 1a). Liposomes showed stability after 100 nm and 200 nm extrusion, with expected hydrodynamic diameters relative to the starting nanoparticle size, on dynamic light scattering (DLS) analysis (Fig. 1b). LiTMX showed reliable uptake of multiple concentrations of 4-hydroxytamoxifen and stability measured at 12 weeks post-production (Fig. 1c).

**LiTMX application to articular cartilage**. Compared with the local application of non-liposomal-packaged 4-hydroxytamoxifen, which results in non-specific cell labeling of the bone marrow compartment when applied to the articular plate in a Gli-CreER[T2]::Rosa26-mTmG mouse model (Fig. 1d, e), topical application of LiTMX application yields precise labeling of cells along the joint surface (Fig. 1f) permitting specific visualization of clonal cell populations. The Gli1 promotor labels bone cells allowing detailed study of bone homeostasis and healing/remodeling after injury. In the mTmG mouse model, all cells express membrane tdTomato (mTomato), but after CreER[T2] recombination, induced cells express membrane green fluorescent protein (mGFP) (Fig. 1d), allowing the activity of specific cell types to be traced. We labeled cells with LiTMX at the articular cartilage at the time of limb injury in mice with the same construct, and were able to visualize (Fig. 1g) and quantify (Fig. 1h) the clonal expansion of cells involved in tissue repair.

Rainbow mice have been described previously for lineage tracing of individual cells with more precision than prior systems permit[8]. This reporter strain contains a multicolor Cre-dependent reporter construct in the ROSA locus (Rosa26-VT2/GK3). With

activation of Cre recombinase, individual cells randomly and permanently express one of four fluorescent protein (FP) colors (cytoplasmic green—eGFP, membrane red—mCherry, membrane yellow—mOrange, and membrane blue—mCerulean) (Fig. 1i), allowing clonal expansion of cells of interest (depending on the Cre driver selected) to be visualized as continuous regions of single colors[8,9].

We induced the articular cartilage with LiTMX in Actin-CreER[T2]::Rosa26-VT2/GK3 mice to achieve focal CreER[T2] induction (Fig. 1j). Compared with standard systemic administration of tamoxifen in corn oil via intraperitoneal (IP) injection for CreER[T2] recombination, which non-specifically labels the entire bone marrow compartment (Fig. 1k), LiTMX showed targeted labeling of tissue-resident cells along the articular surface (Fig. 1l) permitting accurate lineage tracing.

**LiTMX for clonal analysis in intact skin and healing wounds**. We applied LiTMX to the intact, dorsal skin of Rainbow mice, with resulting visualization of clonal cell expansion on 2D analysis (Supplementary Fig. 1a–b). We next created dorsal, full-thickness skin wounds with splinting of the wound edges using silicone washers in the manner described by Galiano et al., 2004, which mimics human wound healing kinetics[10], in Actin-CreER[T2]::Rosa26-VT2/GK3 Rainbow mice (Fig. 2a, b). Unlike systemic tamoxifen induction (delivered IP or via oral gavage (OG)), LiTMX only activates the Rainbow construct (Fig. 2a) locally demonstrated by the absence of Rainbow cells in the peripheral blood (Fig. 2c). Application of LiTMX reliably activates the Rainbow construct with near equal distribution of each of the fluorescent colors across the wound bed (Fig. 2d).

With topical application of LiTMX, in vivo clonal analysis of cutaneous wound healing is visualized on 2D confocal imaging of sectioned specimens at post-operative day (POD) 9 (Fig. 2e), with the size of clones increasing at POD 12 after wounding (Fig. 2f), and further at the time of wound closure, POD 15 (Fig. 2g). The resident cells in the wound appear labeled at POD 9 when LiTMX induction is conducted at the time of wounding, but the systemic inflammatory infiltrate recruited to the wound appears green (GFP+) only indicating that those cells have not been induced (Fig. 2e, left panel, white box inset). Increased incidence of fluorescent cell labeling is seen with increased LiTMX dose (Fig. 2e, right panel).

To visualize the 3D architecture of these wound-responsive cells, we developed a whole-mount technique using the cutaneous wound model (Fig. 3a–c). Histological staining of full-thickness skin wounds from the same model at POD 15, revealed linear migration of cells involved in wound healing (Fig. 3d). Clonal expansion of tissue-resident (non-circulating) wound-responsive cells in the wound bed can be visualized in 3D on Imaris analysis (Fig. 3e). Clones of cells involved in tissue regeneration increase in size over time during the healing process (Fig. 3f). After LiTMX application, robust Rainbow construct activation is particularly apparent along the wound edges, with linear expansion of distinct clones from activated progenitor cells (Fig. 3g—POD 10) engaged in tissue regeneration, seen on whole-mount confocal imaging (Fig. 3h—POD 14, Fig. 3i—POD 21). Clonality is maintained in the dermis, as well as in the induced epidermis using the Actin-CreER[T2]::Rosa26-VT2/GK3 Rainbow mouse mode, over the long term after 2 months of healing and tissue remodeling (Supplementary Fig. 2h–i).

**LiTMX compared with standard techniques**. To further validate our methodology, we produced LiTMX at various concentrations

(10 mg/ml, 20 mg/ml, and 30 mg/ml) and applied these to the femoral periosteum in *Gli-CreER^{T2}::Rosa26-mTmG* mice. Consistent local induction of the mTmG construct along the periosteal surface resulted, and increased dosage correlated with increased frequency of labeled cells (Supplementary Fig. 2a–c). Standard techniques for CreER^{T2} recombinase induction, systemic administration of tamoxifen in corn oil via IP injection, or OG at both low and high doses, yield patchy patterns of cell recombination in complex tissue such as skin and connective tissue (Supplementary Fig. 2d–g). In contrast, LiTMX application

permits more precise cellular labeling of a specific cell lineage of interest (Fig. 2e–g, Fig. 3g–i).

**Clonal analysis in adipose-associated vasculogenic cells**. LiTMX application is beneficial for tracing vasculogenic lineages and can be used for long-term clonal analysis. Adipose tissue engaged with vasculature in the inguinal fat pad of Rainbow mice shows highly clonal tissue at 6 months after labeling (Fig. 4a). With LiTMX activation at 1 week, individual cell labeling has occurred

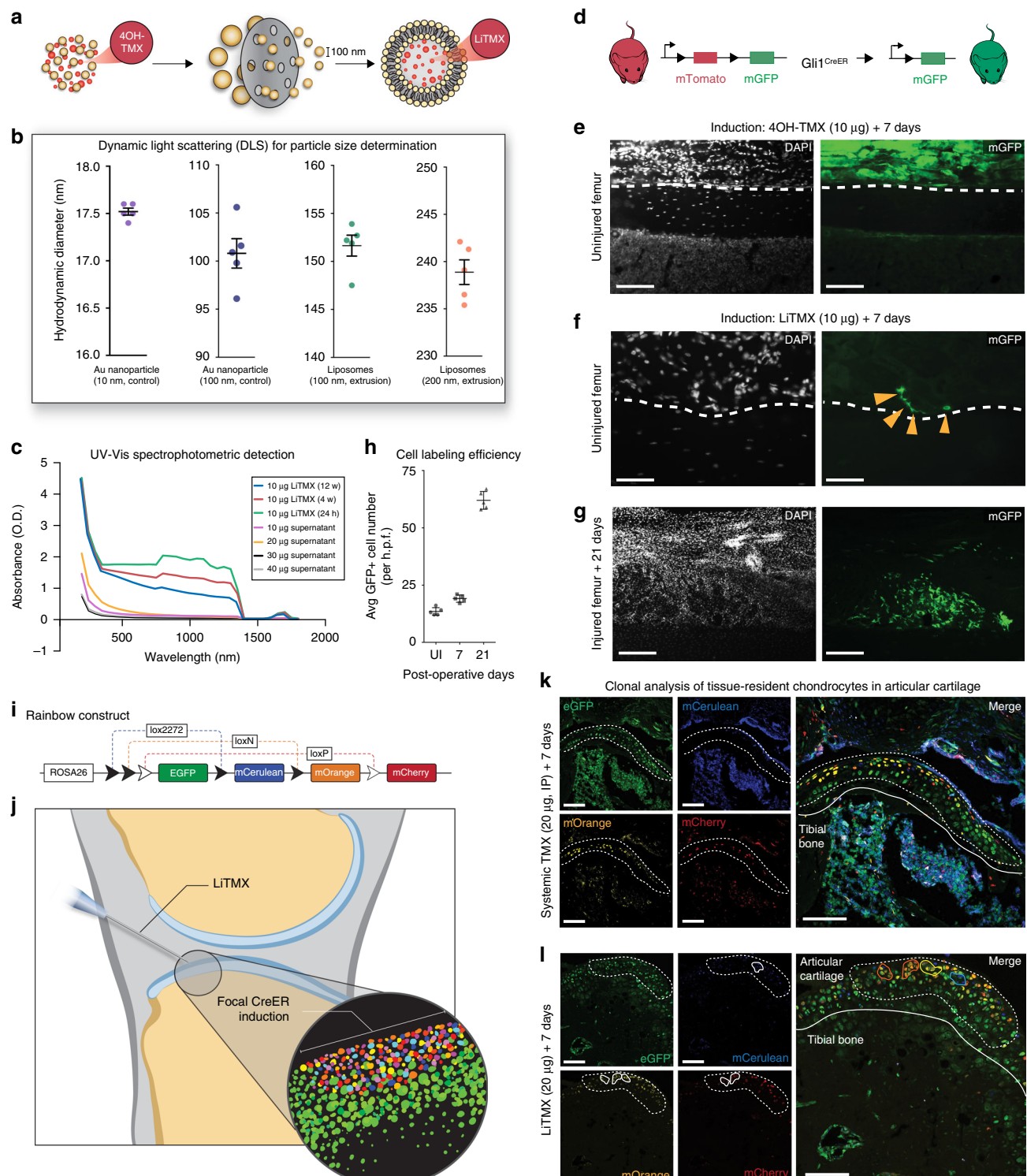

**Fig. 1** Generation of LiTMX for targeted tissue labeling. **a** Schematic of LiTMX generation and extrusion to 100 nm diameter. **b** Liposomal stability as determined by DLS. Hydrodynamic diameter after filtered extrusion of gold (Au) control nanoparticles at 10-nm core size (purple) and 100-nm core size (blue), and for liposomes at 100 nm (green) and 200 nm extrusion (orange). Experiment conducted with $n = 5$ replicates per condition. **c** Stability of LiTMX at 24 h (green), maintained at 6 (red) and 12 weeks (blue) after production, using UV-Vis spectrophometric detection. Following production of LiTMX at various concentrations (10 μg—purple, 20 μg—orange, 30 μg—black, 40 μg—gray), the supernatants show uniformly minimal tamoxifen content demonstrating near complete uptake by the liposomes. **d** Cre-driven recombination resulting in GFP expression in the bone marrow compartment. **e** Local application of non-liposomal 4-hydroxytamoxifen (10 μg) to the femoral periosteum (white dotted lines indicate the superficial periosteum and femur cortical bone border, left panels—white indicates DAPI, right panels—green indicates GFP). **f** In comparison to (**e**), with application of LiTMX at the same dose (individual GFP+ cells indicated by yellow arrowheads). **g** Twenty-one days after limb injury, clonal expansion of labeled cells involved in healing occurs along the bone periosteal surface. **h** The average size of GFP+ clones from **g** increases with time post-injury (UI = uninjured, $n = 5$ replicates per timepoint, error bars indicate standard deviation, SD). **i** Schematic of the Rainbow mouse construct. **j** Schematic of CreER-driven Rainbow construct induction by LiTMX localized to the articular cartilage. Inset shows whole mount of Rainbow labeled tissue. **k** Systemically induced (20 μg, IP) resident chondrocytes of the articular cartilage. (Individual channels at left, merged at right, dotted white line indicates articular cartilage, solid white line indicates tibial bone edge, and solid colored lines indicate individual cell clones). **l** Locally labeled (20 μg) articular cartilage resident chondrocytes by LiTMX in **k**. Experiments conducted with $n = 3$ replicates per timepoint (where applicable) per condition (unless otherwise indicated), error bars denote standard error of the mean (s.e.m., unless otherwise indicated), scale bars represent 200 μm (unless otherwise indicated)

(Fig. 4b, middle panel). Clonal expansion of the labeled cells increases over time (Fig. 4b, right panel) in comparison to no tamoxifen activation (Fig. 4b, left panel).

**LiTMX for lineage tracing with tissue-specific Cre drivers**. We applied LiTMX to soft tissue in mice possessing more specific Cre drivers, *PDGFRα-CreER^T2::Rosa26-mTmG* and *αSMA-CreER^T2::Rosa26-mTmG*, which mark dermal fibroblasts and myofibroblasts, respectfully (Fig. 4c–i). When applied to intact, dorsal skin in *PDGFRα-CreER^T2::Rosa26-mTmG*, the contribution of cells expressing this surface marker were highlighted (Fig. 4d). After skin wounding, clonal expansion of this fibrogenic cell populations occurs after 14 days of healing (Fig. 4e) and 21 days of healing (Fig. 4f). When LiTMX is applied to intact, dorsal skin in *αSMA-CreER^T2::Rosa26-mTmG* mice, cells engaged with vasculature were labeled (i.e., pericytes or smooth muscle cells around blood vessels) (Fig. 4g). After wounding, expansion of these cells was seen in the wound stroma at the same time points (Fig. 4h–i).

## Discussion

The described technique permits identification of clonal cell populations, which can be traced and imaged as described, isolated via laser-capture micro-dissection, or subjected to fluorescence-activated cell sorting (FACS) for subsequent molecular profiling or in vitro culture. Our methodology is highly relevant to living tissue.

A specific advantage of this system includes the effectiveness of local LiTMX application to the tissue surface, which can offer distinct benefits at sites tissue delicacy. Our system has several potential limitations. Primarily, we document successful application of this technique to distinct tissue types, however, we do not know if liposomal 4-hydroxytamoxifen induction of the mTmG, Rainbow, or other constructs of interest would be effective in all areas of the body. In addition, we have carefully determined the optimal dosage and timing of tamoxifen treatment to achieve reliable Rainbow activation for the tissues examined, but again this may not be standard across all tissues and could require additional experimentation. Finally, compared with systemic tamoxifen administration, which can be provided to a mouse in a non-invasive manner (via IP or OG), delivery of LiTMX can necessitate a procedure to expose certain target cells such as the tibia or peritoneum. Compared with systemic tamoxifen administration, however, LiTMX provides the opportunity to induce specific tissues with significantly more precision and efficiency.

In summary, we have successfully generated 4-hydroxytamoxifen liposomal vesicles for targeted induction of CreER^T2-mediated recombination. This technology is efficient and reproducible in a

variety of animal tissue types at various concentrations. Spatial and temporal control over Cre driver activation presents a significant advancement providing a tool for multimodal delineation of tissue-specific processes. This methodology is feasible and accurate to conduct cellular lineage tracing in vivo in dynamic tissues with great accuracy and specificity. The models discussed demonstrate a unique opportunity to locally lineage trace cells involved in tissue regeneration. This methodology is highly applicable to developmental biology, and tissue repair and regeneration research, as well as projects involving skeletal, hematologic, and cancer biology.

## Methods

**Animals**. *Actin-CreER^T2*, *αSMA-CreER^T2*, *PDGFRα-CreER^T2*, and *Gli1-CreER^T2* mouse strains were obtained from Jackson Laboratories, *Rosa26-VT2/GK3* Rainbow mice were provided as a gift from the Weissman Laboratory, Stanford University School of Medicine. All animal procedures were carried out under the guidance of the Stanford University Administrative Panel on Laboratory Animal Care (APLAC).

Prior to induction, mTmG (*Actin-CreER^T2::Rosa26-mTmG, αSMA-CreER^T2::Rosa26-mTmG, Gli1-CreER^T2::Rosa26-mTmG, and PDGFRα-CreER^T2::Rosa26-mTmG*) and Rainbow mice (*Actin-CreER^T2::Rosa26-VT2/GK3*) were assessed for Cre leakiness in multiple tissue types including skin, bone, and large intestine.

**Tamoxifen preparation for systemic induction**. Prior to preparing tamoxifen (Sigma-Aldrich), corn oil (Sigma-Aldrich) was sterilized using a 0.22 μM syringe filter (Millipore Sigma). Tamoxifen was dissolved at a concentration of 20 mg/ml, unless otherwise specified, wrapped in foil and placed in a 37 °C oven for 24 h. After 24 h, the tamoxifen was vortexed at 30 s intervals to ensure the tamoxifen was completely dissolved in the corn oil. Aliquots of the tamoxifen were stored at −20 °C for long-term use.

**4-Hydroxytamoxifen preparation for localized induction**. 4-Hydroxytamoxifen powder (Sigma-Aldrich) was resuspended in 100% methanol per the manufacturer's recommendations. A dose of 10 μg, unless otherwise specified, was applied to the site of interest topically using a pipette.

**Intraperitoneal injection of tamoxifen**. Prior to injection, the tamoxifen was thawed and vortexed to ensure that no precipitate was present within the solution. Mice were anesthetized with 2% inhaled isoflurane (Henry Schein, Melville, NY). Using a 1-ml syringe (BD) attached to a 25 gauge needle (McKesson), the tamoxifen was intraperitoneally administered at a dose of 20 μg, unless otherwise specified, for a course of 5 days.

**Oral gavage administration of tamoxifen**. Prior to injection, the tamoxifen was thawed and vortexed to ensure that no precipitate was present within the solution. Mice were scruffed by hand and held in an upright position. Using a 1-ml syringe (BD) attached to a 20 gauge curved oral gavage needle (McKesson), the tamoxifen was administered at a dose of 20 μg, unless otherwise specified, for a course of 5 days.

**Generating liposome vesicles**. In a 25-ml round bottom flask, 14 μmol of a 90:10 mol:mol mixture of DMPC (1,2-dimyristoyl-*sn*-glycero-3-phosphocholine) and cholesterol (Avanti Polar Lipids) was desiccated under a 5 kPa stream of nitrogen gas

                                                                          

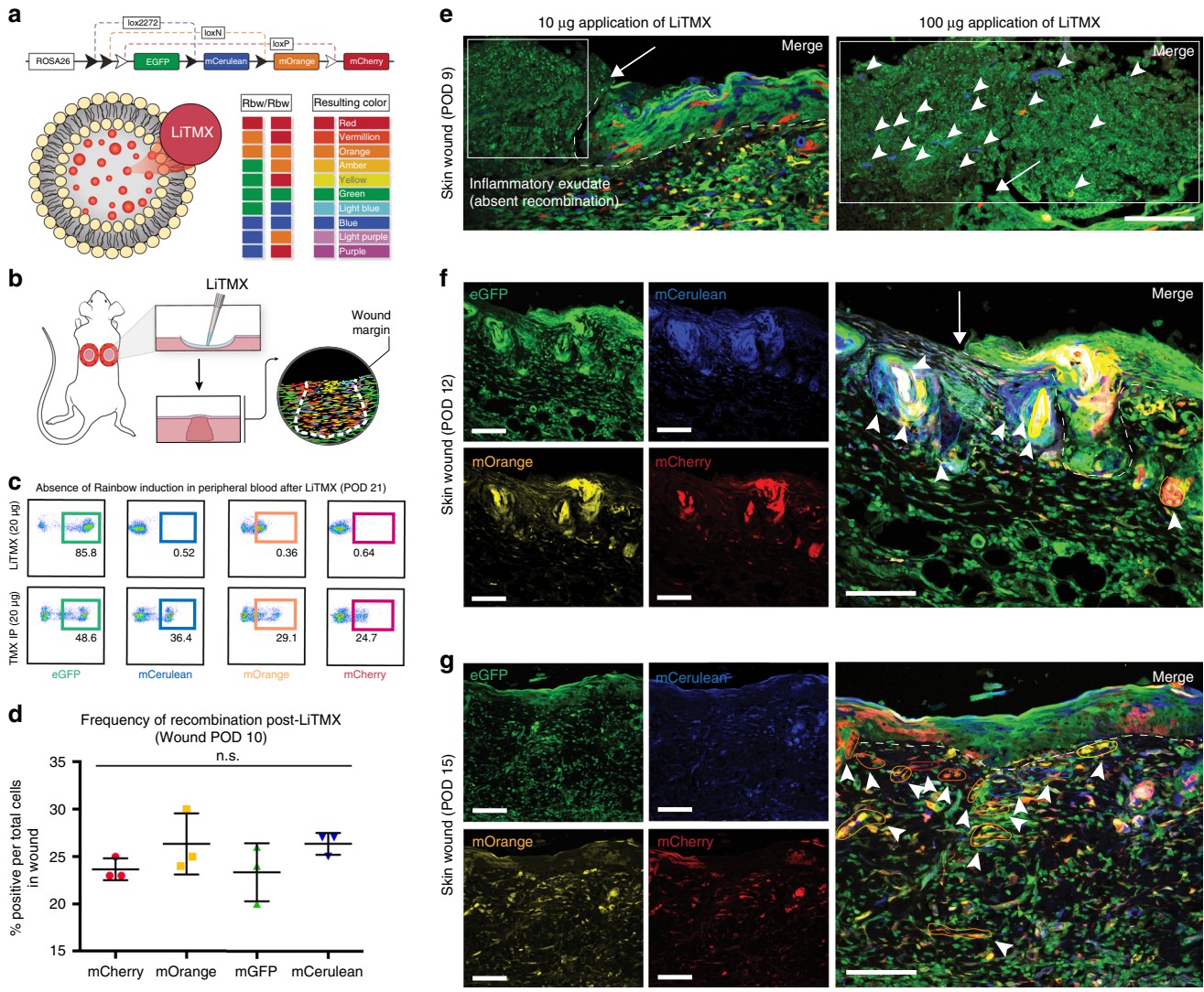

**Fig. 2** Application of LiTMX labeled cells in wound healing. **a** Schematic of the Rainbow reporter mouse construct showing random Cre-mediated recombination after LiTMX induction and resultant cellular color code. Mice homozygous for the Rainbow construct harbor the potential for 10 different colors after recombination. **b** Rainbow cell labeling in wound healing is achieved via local application of LiTMX using a splinted, dorsal, full-thickness excisional wound model. **c** Fluorescent activated cell sorting (FACS) analysis of peripheral blood from mice treated with LiTMX (20 μg) (top panels) versus systemic tamoxifen administrated IP (20 μg) (bottom panels). **d** Randomization of color-coding in activated Rainbow wound tissue shows equal distribution of each fluorescent-labeled cell color (mCherry, mOrange, eGFP, mCerulean) at POD 10 after wounding. **e** With topical application of LiTMX, in vivo clonal analysis of cutaneous wound healing is visualized at POD 9 on confocal imaging of 2D sectioned specimens (left panel: ×20, white arrow indicates wound edge, white dotted line indicates dermo-epidermal junction of skin, white inset box indicates systemic inflammatory infiltrate with absent recombination; right panel: ×40, close up of area indicated by white arrow in left panel, arrowheads indicate increased incidence of fluorescent cell labeling with increased LiTMX dose). **f** Labeled clone size increases at POD 12. (Individual channels at left, merged at right, white arrow indicates wound edge, white dotted line indicates dermo-epidermal junction of skin, solid colored lines highlight individual colored cell clones). **g** Clone size continues to increase at time of wound closure, POD 15. Experiments conducted with $n = 3$ replicates per timepoint (where applicable) per condition (unless otherwise indicated), 2 dorsal wounds per mouse, error bars denote standard error of the mean (s.e.m.), scale bars represent 200 μm (unless otherwise indicated)

for 5 min, followed by vacuum desiccation at 25 °C for 3 h. Then 935 μl of 1 × PBS (Fisher BioReagents, 10 × stock solution—diluted to 1 × with DI water) was added to the desiccated lipids, followed by sonication (Branson) in a water bath for 15 s. The lipids were then reconstituted by extruding 35 times through a 100-nm pore-diameter polycarbonate membrane (Sigma-Aldrich) at 32 °C.

**Characterizing liposome size**. The liposomes were spun down at $30,230 \times g$ (13,000 r.p.m.) for 10 min using a micro benchtop centrifuge. The liposomes were resuspended by vortexing in PBS. Next, using a NanoBrook Omni dynamic light scattering (DLS) instrument (Brookhaven), DLS measurement was performed using a 10 s acquisition time at 37 °C (with the laser power adjusted to maintain the intensity between 500,000 and 2,000,000 counts). The radii, size distribution, as well as the poly dispersity index (PDI) of the liposomes were calculated using the regularization algorithm provided by the associated software. We found that the

100-nm extrusion method yielded homogeneous liposomes with a mean diameter of 151.2 ± 1.9 nm, while extrusion through a 200-nm membrane yielded a mean diameter of 242.1 ± 2.07 nm.

**Preparing LiTMX**. The liposome vesicles were incubated with 4-hydroxytamoxifen (Sigma-Aldrich) in a ratio of 10 mg to 1 ml (10 μg; based on wet mass after centrifugation), 20 mg to 1 ml (20 μg), 30 mg to 1 ml (30 μg), or 40 mg to 1 ml (40 μg) under nitrogen gas at 25 °C for 6 h. The supernatant was decanted and the liposomal pellets were resuspended with an equal volume of 1 × PBS and stored under nitrogen gas at 4 °C. The loading capacity and efficiency of the 4-hydroxytamoxifen in the liposome vesicles was calculated, as well as the retention profile for 4-hydroxytamoxifen based on the different ratios. Optimal concentration of LiTMX was determined in different tissue compartments of different mice strains prior to performing the experiments.

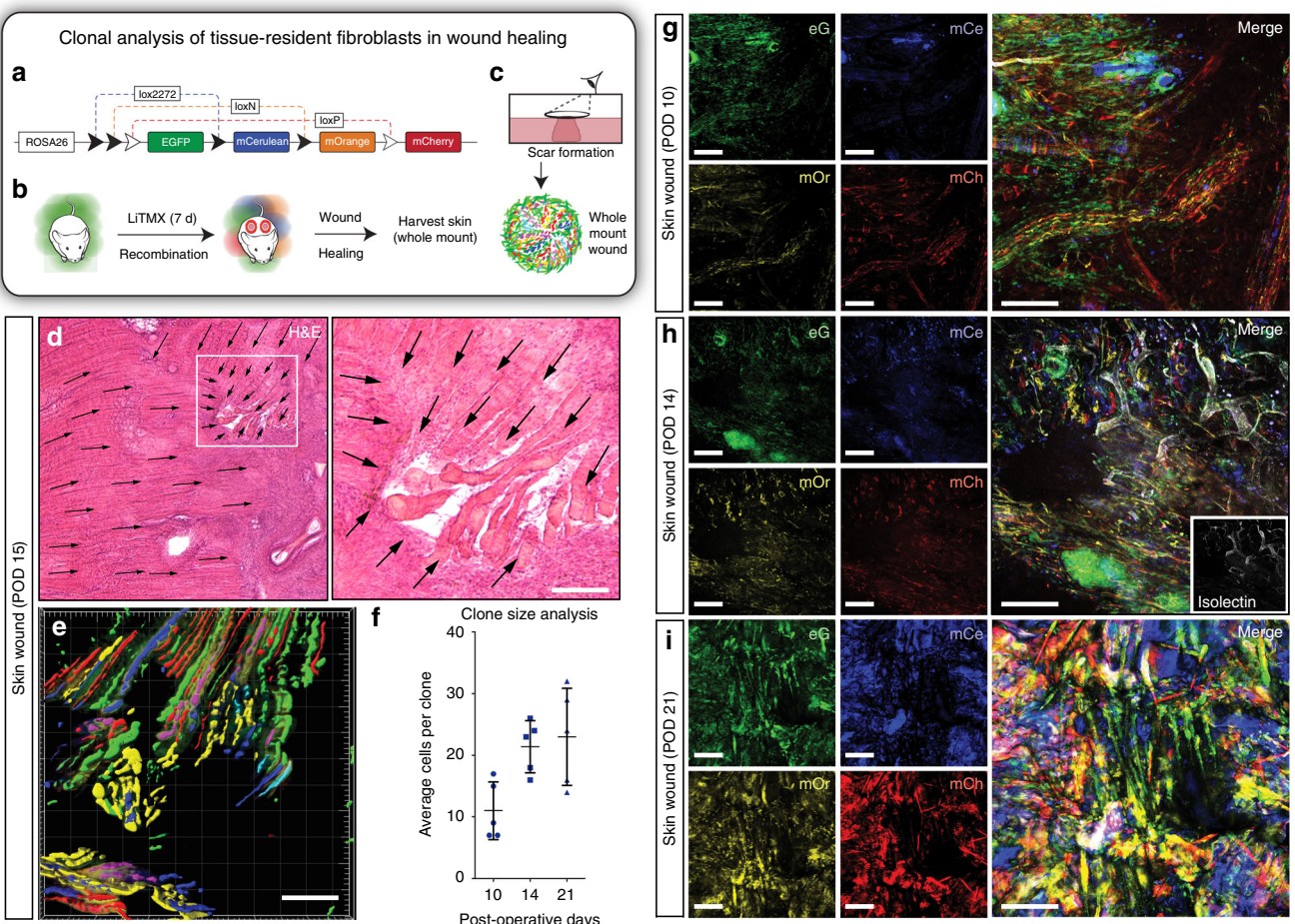

**Fig. 3** LiTMX application in Rainbow mice for clonal analysis in wound healing. **a** Schematic of Cre-mediated recombination and resultant cellular color code change in Rainbow mice. **b** Schematic shows splinted, full-thickness excisional skin wounds with local application of LiTMX. **c** Healing cutaneous wounds are excised, whole mounted and 3D confocal imaging is conducted. **d** A ×10 magnification of H&E staining of a dorsal wound on cross-section (left), white solid box magnified at ×20 reveals linear migration of cells involved in wound healing (right) (black arrows indicate direction of cell migration). **e** Whole-mount wound at POD 15 visualized using Imaris software shows linear migration from the wound edge towards the center of fluorescent-labeled clones of cells involved in wound healing. **f** The average number of cells per clone increases with time post-wounding (data quantified from **g–i**, $n = 5$ replicates per timepoint, error bars indicate SD). **g** After local LiTMX dosing at time of cutaneous wounding in Rainbow mice, confocal imaging of whole-mount specimens shows clonal outgrowth of tissue-resident (non-circulating) wound-responsive cells at POD 10. (eG = eGFP, mCe = mCerulean, mOr = mOrange, and mCh = mCherry, individual channels at left, merged images at right). **h** Clone size increases at POD 14 (white staining, inset, indicates isolectin staining to highlight vasculature). **i** Clone size continues to increase during remodeling after the wound is healed, at POD 21. Experiments conducted with $n = 3$ replicates per timepoint (where applicable) per condition (unless otherwise indicated), 2 dorsal wounds per mouse, scale bars represent 200 μm (unless otherwise indicated)

**Topical application of LiTMX to unwounded mouse skin**. Adult mTmG (*Actin-CreER^T2::Rosa26-mTmG*, *αSMA-CreER^T2::Rosa26-mTmG*, and *PDGFRα-CreER^T2:: Rosa26-mTmG*) and Rainbow (*Actin-CreER^T2::Rosa26-VT2/GK3*) mice were anesthetized with 2% inhaled Isoflurane (Henry Schein, Melville, NY) and 30 mg/ kg Buprenorphine SR (Reckitt Benckiser Pharmaceuticals Inc., Richmond, VA) was injected subcutaneously for pain control. Hair on the dorsal skin was removed with hair clippers followed by depilatory cream (Nair™, Church & Dwight Co., Ewing, NJ). Silicone, 12-mm diameter washers (Invitrogen) were secured around the perimeter of the regions of interest with cyanoacrylate glue and interrupted 6-0 nylon sutures (Ethilon®, Inc., Somerville, NJ, purchased from eSutures.com, Mokena, IL) to adhere the washers to the skin. LiTMX (20 μg, unless otherwise specified) was delivered topically to each of the 4 quadrants (1 μl per quadrant) at the inner diameter of the ring using a 5 μl Hamilton syringe (Hamilton Company, Reno, NV), for a course of 2 days. After 10 min, a sterile Tegaderm dressing (3M Healthcare, St Paul, MN) was applied to the area to protect the skin within the ring. Animals were euthanized and intact skin was harvested 1 week after induction.

**Application to splinted full-thickness excisional wounds**. Adult mTmG (*Actin-CreER^T2::Rosa26-mTmG*, *αSMA-CreER^T2::Rosa26-mTmG*, and *PDGFRα-CreER^T2:: Rosa26-mTmG*) and Rainbow (*Actin-CreER^T2::Rosa26-VT2/GK3*) mice were anesthetized and hair on the dorsal skin was removed as previously described[9]. The

skin was prepared for aseptic surgery with alternating alcohol and Betadyne prep (three times each) to the site followed by placement of a sterile drape. A 6-mm full-thickness wound was created extending through the panniculus carnosus on the dorsum of the mice. A silicone washer was secured around the perimeter of the wound and LiTMX (20 μg, unless otherwise specified) was delivered topically to each of 4 quadrants in the wound followed by placement of a sterile dressing, for a course of 2 days. The dressing was changed every other day until the wound was closed (defined as the time at which the wound bed was completely re-epithelialized and filled with new tissue). Digital photographs were taken at the time of surgery and every other day at the time of dressing changes until the wound was closed. Animals were euthanized and the wound and surrounding skin was harvested at various time points after induction.

**Application to the inguinal fat pad**. Rainbow (*Actin-CreER^T2::Rosa26-VT2/GK3*) mice were placed under anesthesia and analgesia as described and placed in the supine position. A small incision was made in the lower abdominal wall and extended through the peritoneum, the inguinal fat pad was accessed and 2 μl LiTMX (20 μg, unless otherwise specified) was delivered topically to the fat pad bilaterally, single dose only. The peritoneum was closed with running 4-0 monocryl and horizontal mattress suture using 6-0 nylon was used to close the skin.

**Application to long bone**. *Gli1-CreER^{T2}::Rosa26-mTmG* and Rainbow (*Actin-CreER^{T2}::Rosa26-VT2/GK3*) mice were placed under anesthesia and analgesia as described and placed in the supine position. A longitudinal incision was made in the skin parallel to the long bone, subcutaneous fat and muscles were divided parallel to muscle fibers. Dose of 10 µg, unless otherwise specified, of 4-hydroxytamoxifen or LiTMX was applied to the surface of the periosteum using a

2.0-µl Hamilton Modified Microliter Syringe (Sigma-Aldrich) at a recorded distance from the tibial tuberosity, taking care not to damage the periosteum, single dose only. Skin was subsequently closed and the mice monitored postoperatively according to protocol. After a 1-week period, the mice received skeletal injuries as specified. Results of LiTMX assays were tested and confirmed in a minimum of three independent experiments.

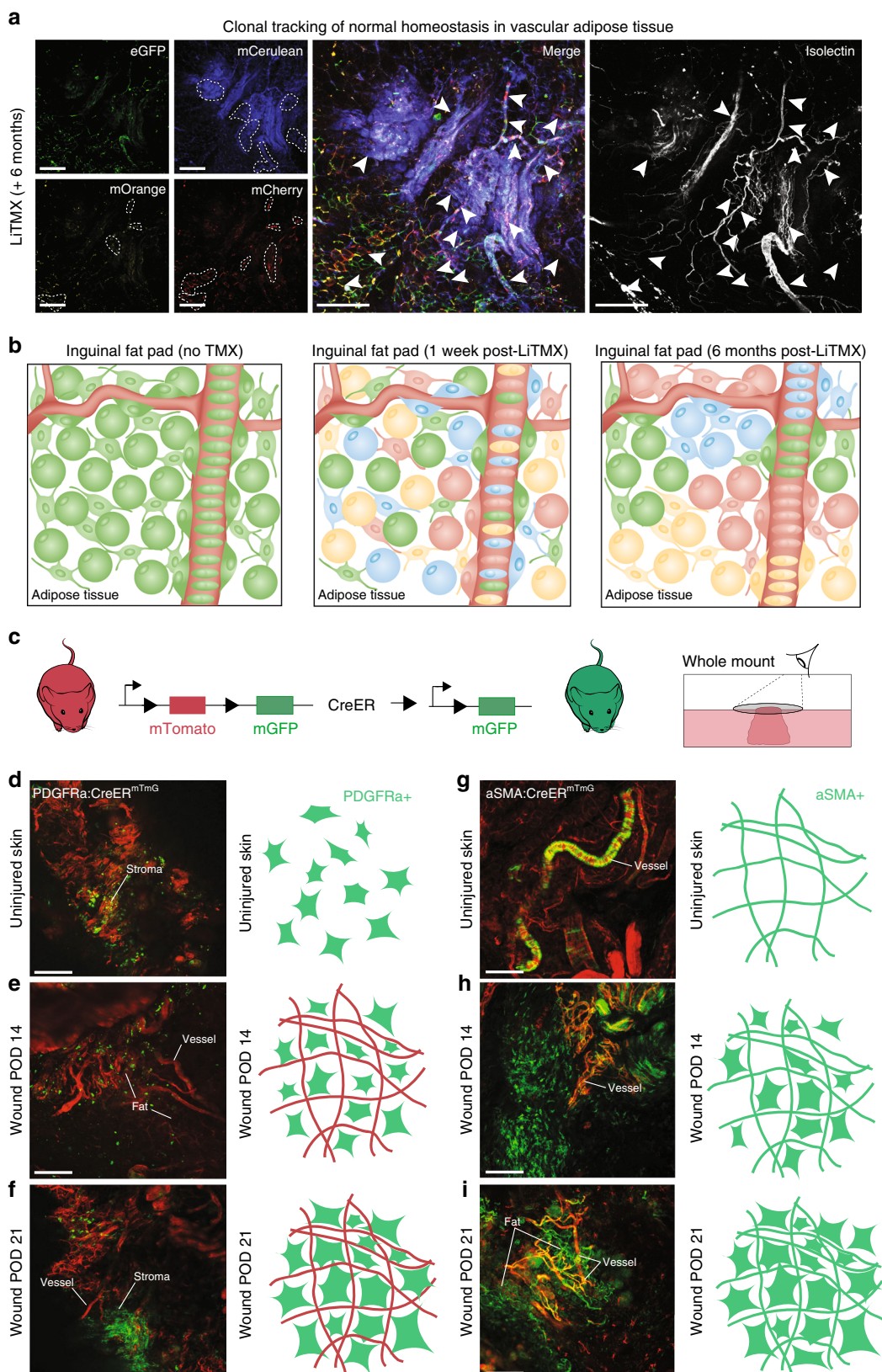

**Fig. 4** LiTMX application for tracing vasculogenic lineages. **a** Clonal tracking of adipose tissue involved with vasculature in the inguinal fat pad in Rainbow mice at 6 months after LiTMX application (left panel shows split channels, middle panel—merged). Isolectin stains cells engaged with vasculature (right panel). White dotted lines (left) and arrowheads (middle and right) mark clonal cell populations. **b** Schematic illustrating the fat pad prior to LiTMX application (left panel), 1 week after application with LiTMX-mediated recombination labeling individual cells (middle panel), and the expansion of clonal populations in uninjured fat pad tissue 6 months after induction (right panel). **c–i** Schematic of experimental application of LiTMX to uninjured skin in *PDGFRα-CreER^T2::Rosa26-mTmG* (left panels, **d–f**) and *αSMA-CreER^T2::Rosa26-mTmG* (right panels, **g–i**) mice: **c** Schematic of the mTmG mouse construct. **d**, **g** LiTMX labeling results in highly-specific localized activation of cells engaged with fibrous tissue and vasculature, respectively, in uninjured skin at homeostasis. (White labeling of wound structures as indicated, confocal images at left in each panel, schematics at right highlight induced cellular phenotypes based on the Cre drivers used). **e**, **h** At POD 14 after cutaneous wounding and LiTMX induction, the *PDGFRα-CreER^T2::Rosa26-mTmG* construct highlights activated fibrogenic cells involved in wound healing, while the *αSMA-CreER^T2::Rosa26-mTmG* construct highlights stromal cells associated with vasculature. **f**, **i** At POD 21 after cutaneous wounding, the cells seen at POD 14 (**e**, **h**), clonally expand. Experiments conducted with $n = 3$ replicates per timepoint (where applicable) per condition (unless otherwise indicated), 2 dorsal wounds per mouse (where applicable), scale bars represent 200 μm (unless otherwise indicated)

**Application to uninjured articular cartilage of the knee**. Rainbow (*Actin-CreER^T2::Rosa26-VT2/GK3*) mice were anesthetized, a 3-mm longitudinal incision over the distal patella extending to the proximal tibial plateau was made. The joint capsule, medial to the patellar tendon was incised. A blunt dissection of the fat pad over the intercondylar area was made, exposing the medial meniscus. LiTMX (20 μg, unless otherwise specified) was topically applied to the medial condyle of the tibial plateau using a 5-μl Hamilton syringe, for a single dose only. The joint capsule was closed with a continuous 8-0 Vicryl® suture with a tapered needle and the subcutaneous layer with 7-0 Vicryl® suture with a cutting needle. The skin was closed the with interrupted 6-0 nylon sutures.

All experiments were conducted with $n = 3$ mice per condition (2 wounds per mouse where applicable). All animal procedures were carried out under the guidance of the Stanford University Administrative Panel on Laboratory Animal Care (APLAC).

**Tissue processing**. For fixation, the samples were placed in 2% paraformaldehyde for 12 h at 4 °C, followed by 3 times 1× PBS washes for 30 min each. At this point, cartilage and bone specimens were decalcified in EDTA for 2 weeks. For 2D sectioning, the samples were then soaked in 30% sucrose in 1× PBS at 4 °C for 24 h. The samples were then incubated in Tissue Tek O.C.T for 24 h at 4 °C followed by freezing on dry ice. The frozen blocks were mounted on a MicroM HM550 cryostat, sectioned at 10 microns thick and transferred to Superfrost/Plus adhesive slides. For 3D imaging, after fixation, whole tissues specimens were mounted on glass slides in Southern Biotech mounting media.

**Imaging**. Using a Leica WLL TCS SP8 Confocal Laser Scanning Microscope, we imaged the 10 micron cryosections and whole tissue mounts with ×20, ×40, and ×63 oil objectives. Precise excitation and hybrid detection of mCerulean, eGFP, mOrange, and mCherry was captured and documented. Z-stacked confocal images were rendered into 3-dimensions for further analysis.

**Data analysis**. ImageJ software was used for analysis of imaging data. Flowjo software was used for FACS analysis of the FACS data. Quantitative analysis was conducted using Graph Pad Prism 6.

**Data availability**. All relevant data are available from the authors.

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

## Acknowledgements

The authors would like to acknowledge Gurpreet Singh and Dan Hunter for their role in the development of liposomal packaging technology. This project was supported by the following funding sources: NIH grant R01DE026730, Hagey Laboratory for Pediatric Regenerative Medicine, Steinhart-Reed Award, Gunn/Olivier fund (M.T.L.), American College of Surgeons resident research award (D.S.F.), and Advanced Residency Training at Stanford (ARTS) award (D.S.F.)

## Author contributions

R.C.R. conceived of the project, conducted data collection and analysis, and prepared the manuscript, D.S.F. and A.S. contributed to data collection and analyses, and preparation of the manuscript, R.E.J., C.D.M., T.L., M.P.M, A.L.M., and C.P.B., all contributed to data collection and analysis, D.C.W contributed to the analyses and guided preparation of the manuscript, M.T.L conceived of the project, oversaw data collection and analyses, and guided preparation and review of the manuscript. All authors reviewed and approved the final manuscript.
