## [Peer Review File · Nature Communications]

REVIEWERS' COMMENTS:

Reviewer #1 (Remarks to the Author):

This manuscript from Ransom, et. al. describes the development of a liposomal system for delivery of 4-hydroxytamoxifen for tightly controlled activation of inducible Cre driver strains, and provides several lineage tracing use case examples. The manuscript is substantially improved with by revisions, particularly with the presentations of comparisons between "standard" tamoxifen delivery and LiTMX. The authors satisfy my major concerns in this area, and it is now clear that the technology is at least similarly efficient, with potentially greater resolution for specific cell type labeling. While it would have been useful to see a comparison to other non-systemic delivery methods (e.g. viral Cre), this wasn't requested in the prior review and given the number of examples, isn't necessary to demonstrate the utility of their method. A few minor editorial/wording modifications are recommended.

- Some parts of the introduction are unclear and/or hard to follow. The sentence on line 35 stating "However, spatial control over induction remains limited" should include "...remains limited to the cell and tissue-specific expression of cre in available driver strains".

- The next sentence "With systemic tamoxifen administration..." is very confusing. Does "gene" refer to the driver or the reporter? The authors should clarify, perhaps removing as the above edit would capture this sentiment.

- The use of "inherent lack" is overstated. Suggest a more neutral statement (with citation) about frequent off-target or non-specific activity.

- "Robust expression of report genes...is not always present". Please provide a citation for this. However, I would advise removing this statement as the LiTMX method would not help with limitations of reporter function.

- The large number of examples results in figures with very small panels that are hard to see (with tiny labeling fonts). Higher resolution figures will be needed for publication as the reader will need to expand them on their screen to read.

- Line 124 should state "To further validate"...

- Lines 159-162. This statement seems to describe the general advantages of in vivo lineage tracing, not anything specific to LiTMX...seems unnecessary.

- Line 164-167: In general, the authors should be careful in their description of what is induced by LiTMX (creERT2 activity) and how it is determined (reporter activity). For this particular sentence, it reads as if Rainbow or mTmG activation in some cell types might be limited, when the real issue is whether LiTMX can be delivered to activate CreERT2. A bit of a semantic argument, but an important distinction as the target Cre expression ITSELF could be the reason the reporter activity is not induced as expected. The point is taken, however, that the general deliverability of LiTMX needs to be established empirically for other tissues and cell types.

Reviewer #2 (Remarks to the Author):

This revised version of the manuscript has adequately addressed all of my comments and criticism that I raised in the previous version. I have no more suggestion for change.

A concise point-by-point response (in red) to the *Nature Communications* reviewer suggestions and editorial requests (in black) is provided as follows.

Reviewer #1 (Remarks to the Author):

This manuscript from Ransom, et. al. describes the development of a lipidosomal system for delivery of 4-hydroxytamoxifen for tightly controlled activation of inducible Cre driver strains, and provides several lineage tracing use case examples. The manuscript is substantially improved with by revisions, particularly with the presentations of comparisons between “standard” tamoxifen delivery and LiTMX. The authors satisfy my major concerns in this area, and it is now clear that the technology is at least similarly efficient, with potentially greater resolution for specific cell type labeling. While it would have been useful to see a comparison to other non-systemic delivery methods (e.g. viral Cre), this wasn’t requested in the prior review and given the number of examples, isn’t necessary to demonstrate the utility of their method. A few minor editorial/wording modifications are recommended.

1. Some parts of the introduction are unclear and/or hard to follow. The sentence on line 35 stating “However, spatial control over induction remains limited” should include “...remains limited to the cell and tissue-specific expression of cre in available driver strains”.

We appreciate this suggestion for revision. The relevant sentence has been revised as suggested by the reviewer.

2. The next sentence “With systemic tamoxifen administration...” is very confusing. Does “gene” refer to the driver or the reporter? The authors should clarify, perhaps removing as the above edit would capture this sentiment.

Thank you for this comment and suggestion for revision. The relevant sentence has been revised for clarity.

3. The use of “inherent lack” is overstated. Suggest a more neutral statement (with citation) about frequent off-target or non-specific activity.

Thank you for this comment. The relevant sentence has been revised as recommended by the reviewer.

4. “Robust expression of report genes...is not always present”. Please provide a citation for this. However, I would advise removing this statement as the LiTMX method would not help with limitations of reporter function.

We appreciate this recommendation. The relevant sentence has been removed as recommended by the reviewer.

5. The large number of examples results in figures with very small panels that are hard to see (with tiny labeling fonts). Higher resolution figures will be needed for publication as the reader will need to expand them on their screen to read.

Thank you for this recommendation. High resolution figures are provided to Nature Communications with our resubmission.

6. Line 124 should state “To further validate”...

We appreciate this recommendation. The relevant sentence has been revised as recommended by the reviewer.

7. Lines 159-162. This statement seems to describe the general advantages of in vivo lineage tracing, not anything specific to LiTMX...seems unnecessary.

Thank you for this recommendation. The relevant sentence has been removed as suggested by the reviewer.

8. Line 164-167: In general, the authors should be careful in their description of what is induced by LiMTX (creERT2 activity) and how it is determined (reporter activity). For this particular sentence, it reads as if Rainbow or mTmG activation in some cell types might be limited, when the real issue is whether LiTMX can be delivered to activate CreERT2. A bit of a semantic argument, but an important distinction as the target Cre expression ITSELF could be the reason the reporter activity is not induced as expected. The point is taken, however, that the general deliverability of LiTMX needs to be established empirically for other tissues and cell types.

Thank you for this comment.

Reviewer #2 (Remarks to the Author):

1. This revised version of the manuscript has adequately addressed all of my comments and criticism that I raised in the previous version. I have no more suggestion for change.

We appreciate this reviewer comment.

EDITORIAL REQUESTS:

There are a number of requirements that need to be addressed. We will be unable to proceed with acceptance of your manuscript until it adheres to these requirements. Please use the tracked changes feature of Microsoft Word to make these changes.

Please provide a point-by-point response to these points with your submission.

Thank you for outlining the editorial revisions required. Our revisions are provided as track changes in the manuscript as requested and a point-by-point response is included with this submission.

* *Nature Communications* uses a transparent peer review system, where for manuscripts submitted from January 2016 we are publishing the reviewer comments to the authors and author rebuttal letters of our research articles online as a supplementary peer review file. Please let us know in the cover letter when submitting the final version of your manuscript if you wish to opt out of this scheme or not. If you are concerned about the release of confidential data, we do permit redactions in the interest of confidentiality. If you would like to remove such information from these files, then please let us know specifically what information you would like to have removed. Please note that we cannot incorporate redactions for other reasons. For more information, please refer to our FAQ page at <https://media.nature.com/full/nature-assets/ncomms/authors/ncomms-transparent-peer-review.pdf>

As noted in our cover letter, we do not make any special request to opt out of this process.

* Your manuscript should comply with our policies and format requirements, detailed in our checklist for authors at:
http://www.nature.com/article-assets/npg/ncomms/authors/ncomms_manuscript_checklist.pdf

The checklist has been re-reviewed and our manuscript made compliant.

* Data availability statements and data citations policy: All *Nature Communications* manuscripts must include a section "Data Availability" at the end of the Methods section or main text (if no Methods). For more information on this policy, and a list of examples, please see <http://www.nature.com/authors/policies/data/data-availability-statements-data-citations.pdf>

- Accession codes for deposited data
- Other unique identifiers (such as DOIs and hyperlinks for any other datasets)
- At a minimum, a statement confirming that all relevant data are available from the authors
- If applicable, a statement regarding data available with restrictions
- If a dataset has a Digital Object Identifier (DOI) as its unique identifier, we strongly encourage including this in the Reference list and citing the dataset in the Data Availability Statement.

Thank you for this explanation. A Data Availability section and the following statement has been added to the end of the Methods section: "All relevant data are available from the authors".

* DATA SOURCES: We strongly encourage authors to deposit all new data associated with the paper in a persistent repository where they can be freely and enduringly accessed. We recommend submitting the data to discipline-specific, community-recognized repositories, where possible and a list of recommended repositories is provided here: <http://www.nature.com/sdata/policies/repositories>

If a community resource is unavailable, data can be submitted to generalist repositories such as figshare (<https://figshare.com/>) or Dryad Digital Repository (<http://datadryad.org/>). Please provide a unique identifier for the data (for example a DOI or a permanent URL) in the data availability statement, if possible. If the repository does not provide identifiers, we encourage authors to supply the search terms that will return the data. For data that have been obtained from publically available sources, please provide a URL and the specific data product name in the data availability statement. Data with a DOI should be further cited in the methods reference section.

Please refer to our data policies here: <http://www.nature.com/authors/policies/availability.html>

Thank you for this recommendation and helpful information.

* To ensure correct hyperlinking of the accession codes in your manuscript, please add the hyperlink or DOI in square brackets directly after the code throughout (for example, '5XRN [<http://dx.doi.org/10.2210/pdb5XRN/pdb>]', '1483958 [<https://dx.doi.org/10.5517/ccdc.csd.cc1lt5m6>]', 'SRP109982 [<https://www.ncbi.nlm.nih.gov/sra/?term=SRP109982>]' or 'NQLW00000000 [https://www.ncbi.nlm.nih.gov/assembly/GCA_002312845.1/]').

Thank you for this information.

Please pay particular attention to the following points, and use the tracked changes feature of

Microsoft Word to make changes to the manuscript:

* Please check whether your manuscript or Supplementary Information contain third-party images, such as figures from the literature, stock photos, clip art or commercial satellite and map data. We strongly discourage the use or adaptation of previously published images, but if this is unavoidable, please request the necessary rights documentation to re-use such material from the relevant copyright holders and return this to us when you submit your revised manuscript.

All of our figures are original – produced solely by the authors of the manuscript, no third-party or adapted images are included in any of our figures.

* Nature journals require authors of life sciences research papers to include relevant details about several elements of experimental and analytical design in their manuscripts. This initiative aims to improve the transparency of reporting and the reproducibility of published results and is described at: <http://www.nature.com/authors/policies/reporting.pdf> To ensure that your manuscript complies with our policy, please pay close attention to the 'methods' and 'legends' sections of our checklist for authors: http://www.nature.com/article-assets/npg/ncomms/authors/ncomms_lifesciences_checklist.pdf

You may also find the following collection of articles on statistics for biologists helpful: <http://www.nature.com/collections/qghhqm>

Additionally, please ensure that an updated editorial policy checklist that verifies compliance with all required editorial policies is completed and uploaded as a Related Manuscript file type with the revised article. Please note that this form is a dynamic 'smart pdf' and must therefore be downloaded and completed in Adobe Reader.

<https://www.nature.com/authors/policies/Policy.pdf>

The checklist has been re-reviewed and our manuscript made compliant.

* The Introduction should be structured in such a way that the background and rationale for the study, including all previous literature, is discussed first, and the current work is discussed only in the final paragraph. Please rearrange the Introduction to adhere to this format: all necessary background and context should be in preceding paragraphs and the major results and conclusions of the current work should be summarised in present tense in the final paragraph. Please ensure adequate references are given so that your work is placed appropriately within the context of the existing literature.

Thank you for this information. Our introduction is structured in this manner. The final paragraph has been revised from the past to present tense as requested.

* Please divide the Results and Methods section into subsections, each with a title of 60 characters or fewer including spaces. Subheadings may not contain punctuation such as commas. Please ensure that a subheading is present at the very beginning of both the Results and Methods sections, even if there is only one subsection.

Thank you for this information. Our Results and Methods sections are structured in this manner.

* We strongly discourage the use of "data not shown" - in each case, please either supply the data as a supplementary figure, or if the statement is not important to the conclusions, remove it.

We appreciate this recommendation. Our single use of this phrase has been removed from the manuscript. The data is not provided as a supplement as it is not important to the conclusions.

* In the Methods, please provide sufficient information such that the experiments could reasonably be reproduced without reference to other papers, and avoid use of the term 'as described previously'.

Thank you for this recommendation. Our Methods section is thorough and provides adequate information such that the experiments could be reproduced without reference to other papers. Aside from in reference to the genetic construct of one of the mouse models used, we avoid use of this term.

* Please avoid using speech marks around words or phrases to convey emphasis.

Thank you for this recommendation. We avoid use of this.

* With regards to the experiments using animal models, please confirm that you have complied with all relevant ethical regulations and that a statement affirming this, and the name of the board and institution that approved the study protocol, is included in the methods section of the manuscript.

Thank you for this reminder. As noted in the Methods section of the manuscript, all animal procedures were carried out under the guidance of the Stanford University Administrative Panel on Laboratory Animal Care.

* Please ensure that centrifugation speeds are stated in xg, not rpm.

Thank you for this information. This information has been revised in the manuscript.

* Please ensure that +/- values are defined at the first point of use within the text and figure legends and numbers of replicates are given.

Thank you for this information.

* In each Figure and Supplementary Figure where error bars are used, they must be defined, and the number of experimental replicates stated. One statement at the end of each figure is sufficient if the error bars are equivalent throughout the figure.

Thank you for this recommendation. This requested information has been provided in the figure legends for each figure.

* Please ensure that at least one micrograph in each equivalent group in each figure is supplied with a representative scale bar, whose length is stated in the corresponding figure legend.

Thank you for this recommendation. At least one micrograph in each equivalent group is supplied in each figure. Scale bars have been added to all images with the length now noted in the figure legends.

* Please remove the scale bar labels from the figures in the main text and Supplementary Information (keeping the scale bar) and incorporate this information in the corresponding figure captions.

Thank you for this note. Scale bar labels have been removed from the figures and the information is now provided in the figure legends instead.

* Please ensure that the corresponding dot plots are overlaid in the bar charts.

Thank you for this recommendation. Our figures include a combination of line, dot and bar charts.

* Figures legends should not exceed 350 words, including titles. Please shorten by removing detailed methodological information and/or interpretation, or, if appropriate, consider splitting the affected figures in two. Note that we allow up to 10 display items (figures and tables) in the main manuscript.

Thank you for this recommendation. Figure legends have been revised to closely reflect this word count limit.

* Please define any new abbreviations, symbols or colours present in your figures in the associated legends, noting that these should be written out in words (blue circles, red dashed line, etc.) as symbols will not appear properly in the HTML text.

Thank you for this recommendation. Abbreviations are defined in the figure legends where applicable.

* Please update any incomplete web links for DOI references and preprints.

Thank you for this recommendation.

* Please supply an acknowledgements section after the Methods section.

Thank you for this suggestion. We have added an Acknowledgements section (track changes in the revised manuscript), which follows the Methods section.

* Please make a statement of competing financial and non-financial interests after the author contributions section that refers to all authors. If there are no competing interests, please add the statement "The authors declare no competing interests."

Thank you for this reminder. Our manuscript includes this section. The wording has been revised to the above.

* Please note that we do not reformat Supplementary Information files; they will be uploaded with the published article as they are submitted with the final version of your manuscript. Please check everything very carefully and remove any track changes from the file. Failure to adhere to our style guidelines will result in delays in production. The only sections we permit in the Supplementary Information file are Supplementary Figures, Supplementary Tables, Supplementary Methods, Supplementary Notes, Supplementary Discussion, Supplementary References. Please supply a single, separate Supplementary Information file.

We appreciate this reminder. Our supplement is now provided as a separate pdf.

* Your paper will be accompanied by a two-sentence editor's summary, of between 250-300

characters, when it is published on our homepage. Could you please approve the draft summary below or provide us with a suitably edited version.

Targeted genetic dissection of tissues can be used to identify cell populations and lineages. Here the authors develop 4-hydroxytamoxifen liposomes for the localised induction of CreERT2."

Thank you. We approve of the above summary draft.

OPEN ACCESS:

Nature Communications is a fully open access journal. Articles are made freely accessible on publication under a CC BY license (Creative Commons Attribution 4.0 International License). This license allows maximum dissemination and re-use of open access materials and is preferred by many research funding bodies.

For further information about article processing charges, open access funding, and advice and support from Nature Research, please visit <http://www.nature.com/ncomms/about/open-access>

SUBMISSION INFORMATION:

In order to accept your paper, we require the following:

* A cover letter describing your response to our editorial requests.

Thank you. We provide a cover letter with this re-submission.

* A separate document detailing your point-by-point response to any issues raised by our referees (please include the referees' comments in this document).

Thank you. We provide a point-by-point response to issues raised by the referees and editors with this re-submission.

* The final version of your text as a Word or TeX/LaTeX file, with any tables prepared using the Table menu in Word or the table environment in TeX/LaTeX and using the 'track changes' feature in Word.

Thank you. We have tracked changes in our revised manuscript document.

* Production-quality versions of all figures, supplied as separate files. To ensure the swift processing of your paper please provide the highest quality, vector format, versions of your images (.ai, .eps, .psd) where available. Please see our brief guide to manuscript submission for further details on the figure formats we can accept. Text and labelling should be in a separate layer to enable editing during the production process. If vector files are not available then please supply the figures in whichever format they were compiled in and not saved as flat .jpeg or .TIFF files. Any chemical structures or schemes contained within figures should additionally be supplied as separate ChemDraw (.cdx) files. If your artwork contains any photographic images, please ensure these are at least 300 dpi.

Thank you. We are re-submitting our files are high-resolution .pdf files.

To ensure that your figures are accessible to colour-blind readers, we encourage you to use alternative colour schemes. For example, rainbow colour scales may be replaced by single-colour intensity scales or greyscale, and red/green image overlays may be replaced with magenta/green. For reference an example of R-script colour blindness palettes can be found here <https://cran.r-project.org/web/packages/iridis/vignettes/intro-to-iridis.html>. Another example for Python can be found here: <http://matplotlib.org/cmocan/>

Whenever possible our data should be accessible to colour-blind readers. However, there may be some limitations in this regard due to the nature of the mTmG and Rainbow mouse constructs.

* The final version of any Supplementary Information (figures, tables, notes etc) in one PDF file. Please add a cover page to the Supplementary Information PDF, including the title of the manuscript and the first author's surname in the format 'Smith et al.' Please submit movies, audio files and data sets as separate files. See <http://www.nature.com/ncomms/submit/how-to-submit#Supplementary-information> for acceptable file formats/sizes.

** Please note that Supplementary Information must be finalised prior to acceptance of the paper. **

Thank you. We provide a separate pdf containing our supplementary information.

* If you wish, an interesting image (but not an illustration or schematic) for consideration as a 'Featured Image' on the Nature Communications homepage. Examples can be seen on our Facebook page: <http://go.nature.com/PGPizM> The file should be 1400x400 pixels in RGB format and should be uploaded as 'Related Manuscript File'. In addition to our home page, we may also use this image (with credit) in other journal-specific promotional material.

Thank you for this opportunity. We have provided a file for consideration based on data presented in our manuscript. We would be thrilled if this was selected as the "Featured Image" for the Nature Communications homepage.

* A completed author checklist, uploaded as a Related Manuscript file type, available at: http://www.nature.com/article-assets/npq/ncomms/authors/ncomms_manuscript_checklist.pdf

Thank you for this information. We provide a completed author checklist as a Related Manuscript file.

* Completed and signed copies of our Multimedia License to Publish (LTP) for any Featured Image suggestions (please use one form for each image and give a scientific description of the image in the 'title' field; do not use "Featured Image" as a title): Multimedia Licence to Publish form

Thank you for this information. We provide a completed form for our suggested "Featured Image".

At acceptance, the corresponding author will be required to complete an Open Access Licence to Publish on behalf of all authors, declare that all required third party permissions have been obtained and provide billing information in order to pay the article-processing charge (APC) via credit card or invoice.

Please note that your paper cannot be sent for typesetting to our production team until we have received these pieces of information; **therefore, please ensure that you have this information ready when submitting the final version of your manuscript.**

<http://mts-ncomms.nature.com/cgi-bin/main.plex?el=A3S3BKCN6A7dJE1I7A9ftdwS1mDdnX4wGg9rpXca3sugZ>

Thank you for this information. We are ready to provide the Open Access Licence to Publish information.

P.S. We recommend that you upload the step-by-step protocols used in this manuscript to the Protocol Exchange. Protocol Exchange is an open online resource that allows researchers to share their detailed experimental know-how. All uploaded protocols are made freely available, assigned DOIs for ease of citation and fully searchable through nature.com. Protocols can be linked to any publications in which they are used and will be linked to from your article. You can also establish a dedicated page to collect all your lab Protocols. By uploading your Protocols to Protocol Exchange, you are enabling researchers to more readily reproduce or adapt the methodology you use, as well as increasing the visibility of your protocols and papers. Upload your Protocols at www.nature.com/protocolexchange/. Further information can be found at www.nature.com/protocolexchange/about.

Thank you for this recommendation. We will consider this opportunity.

P.S. To help the scientific community achieve unambiguous attribution of all scholarly contributions, Nature Communications encourages all authors to create and link an ORCID identifier to their account. Please ensure that all co-authors are aware that they can add their ORCIDs to their accounts, so that it will display on this paper. If they so wish, they must do so before the paper is formally accepted. It will not be possible to add ORCIDs post-acceptance, e.g. at proof. To add an ORCID please follow these instructions:

1. From the home page of the MTS click on 'Modify my Springer Nature account' under 'General tasks'.
2. In the 'Personal profile' tab, click on 'ORCID Create/link an Open Researcher Contributor ID (ORCID)'. This will re-direct you to the ORCID website.
- 3a. If you already have an ORCID account, enter your ORCID email and password and click on 'Authorize' to link your ORCID with your account on the MTS.
- 3b. If you don't yet have an ORCID account, you can easily create one by providing the required information and then clicking on 'Authorize'. This will link your newly created ORCID with your account on the MTS.

Thank you for this recommendation. We will consider this opportunity.